# New Random Aromatic/Aliphatic Copolymers of 2,5-Furandicarboxylic and Camphoric Acids with Tunable Mechanical Properties and Exceptional Gas Barrier Capability for Sustainable Mono-Layered Food Packaging

**DOI:** 10.3390/molecules28104056

**Published:** 2023-05-12

**Authors:** Giulia Guidotti, Michelina Soccio, Massimo Gazzano, Valentina Siracusa, Nadia Lotti

**Affiliations:** 1Department of Civil, Chemical, Environmental, and Materials Engineering, University of Bologna, 40138 Bologna, Italy; giulia.guidotti9@unibo.it (G.G.); nadia.lotti@unibo.it (N.L.); 2Interdepartmental Center for Industrial Research on Advanced Applications in Mechanical Engineering and Materials Technology, CIRI-MAM, University of Bologna, 40136 Bologna, Italy; 3Organic Synthesis and Photoreactivity Institute, ISOF, CNR, 40129 Bologna, Italy; massimo.gazzano@isof.cnr.it; 4Department of Chemical Science, University of Catania, 95125 Catania, Italy; vsiracus@dmfci.unict.it; 5Interdepartmental Center for Agro-Food Research, CIRI-AGRO, University of Bologna, 47521 Cesena, Italy

**Keywords:** 2,5-furandicarboxylic acid, (1R, 3S)-(+)-Camphoric Acid, poly(butylene 2,5-furandicarboxylate), bio-based copolymers, random copolymers, thermal properties, mechanical properties, gas barrier properties

## Abstract

High molecular weight, fully biobased random copolymers of 2,5-furandicarboxylic acid (2,5-FDCA) containing different amounts of (1R, 3S)-(+)-Camphoric Acid (CA) have been successfully synthesized by two-stage melt polycondensation and compression molding in the form of films. The synthesized copolyesters have been first subjected to molecular characterization by nuclear magnetic resonance spectroscopy and gel-permeation chromatography. Afterward, the samples have been characterized from a thermal and structural point of view by means of differential scanning calorimetry, thermogravimetric analysis, and wide-angle X-ray scattering, respectively. Mechanical and barrier properties to oxygen and carbon dioxide were also tested. The results obtained revealed that chemical modification permitted a modulation of the abovementioned properties depending on the amount of camphoric co-units present in the copolymers. The outstanding functional properties promoted by camphor moieties addition could be associated with improved interchain interactions (π-π ring stacking and hydrogen bonds).

## 1. Introduction

Nowadays, packaging, both rigid and flexible, represents the largest plastics market, covering about 44% of the global production, which is around 390.7 Mton/year [1]. This number is predicted to further grow in the next few decades and will exacerbate the problem of environmental pollution if proper waste disposal management is not developed rapidly and correctly. At the same time, the growing focus on waste accumulation and the political and social support for reaching the so-called circular economy are requiring a careful choice of plastic materials for common use [2]. As is known, one of the cornerstones of the circular economy is the minimum environmental impact of a material throughout its entire life cycle, from the design, the choice of raw materials, and the production processes until disposal [3,4]. The synthesis of sustainable plastics from green monomers and the recycling after use are of particular importance to boost the development of circularity in production processes. As to this last, it is not always feasible or economically favorable, as many wastes are contaminated by organic matter and cannot be easily recycled, or recycling costs are higher than those needed to produce virgin plastics [5,6]. This is particularly true when the food-packaging field is considered, where restrictions on contaminants and constituents, which can migrate into food or drinks in quantities harmful for human health, place further limits on the use of recycled material. This is in agreement with Regulation (EC) 1935/2004, which sets general principles of good manufacturing practice and safety for food-contact materials. It has been estimated that in the European Union, only 5% of plastic used for food packaging is currently recycled in a closed loop [2,7]. On the other hand, in recent years, many efforts have been made to search for green monomers for the synthesis of new plastic materials. To achieve this aim, renewable sources as well as sustainable synthetic processes that minimize the use of solvents and the emissions of greenhouse gases are key targets that must be considered. Another initiative to reduce the environmental impact of plastics the substitution of rigid packaging with flexible ones. In fact, in such a way, we can further decrease plastic volumes of production, both of materials and waste, maximizing sustainability [8,9].

A fully bio-based monomer that caught the attention of both academia and industry is 2,5-furandicarboxylic acid (2,5-FDCA), which is characterized by a chemical structure very similar to terephthalic acid, the precursor of PET. Moreover, it belongs to the list of 12 high-value-added chemicals obtained from sugars by the United States Department of Energy [10,11]. Indeed, 2,5-FDCA can be synthesized starting from biomass and sugars containing six carbon atoms (i.e., fructose or glucose) that are dehydrated into hydroxy methyl furfural (HMF), which can be, in turn, oxidized into 2,5-FDCA [12]. The interest in this monomer and in the polymers that derive from it is documented by many studies present in the literature as well as by the industrial production started by many companies all over the world, such as Avantium, Ava Biochem, DuPont, and ADM [13,14,15,16,17]. Poly(butylene 2,5-furandicarboxylate) (PBF) is the green alternative to petroleum-derived poly(butylene terephthalate) PBT, and has recently been the subject of several studies. Indeed, random and block copolymers, blends, and composites have been prepared and investigated [18,19,20,21,22,23]. Considering this scenario, in the present paper, novel *ad-hoc* designed sustainable random copolymers of PBF with mechanical and barrier properties suitable for flexible food packaging have been synthesized, processed in the form of film, and characterized to check the functional requirements.

Another cheap and largely available bio-based building block that is recently arousing particular interest is (1R, 3S)-(+)-Camphoric Acid (CA), a monomer consisting of a five-membered aliphatic ring. This diacid is the oxidation product of bicyclic terpene (1R)-(+)-camphor obtained from the camphor laurel tree, originally present in Borneo, Taiwan, and eastern Africa but now cultivated in many other parts of the world [24,25,26]. An alternative way to produce camphoric acid involves the use of turpentine, obtained by distillation from the resin of blue pine (*Pinus Excelsa*) [27,28]. In 2018, the Fraunhofer Institute for Interfacial Engineering and Biotechnology (IGB) started, in collaboration with scientific and industrial partners from other countries, some research on sustainable processes for biobased camphoric acid production [29]. The high interest in this monomer is also demonstrated by many scientific publications in which CA is used as a building block for the realization of copolymers and composites [30,31,32,33,34,35]. Camphoric acid has been recently utilized by the authors of this paper to chemically modify poly(butylene *trans*-1,4-cyclohexanedicarboxylate), a rigid aliphatic polyester [36].

PBF is not suitable for the realization of flexible packaging because of its mechanical rigidity [13]. Considering copolymerization represents a winning strategy to improve the unsatisfactory characteristics of a material without worsening the already good ones, we decided to introduce camphoric acid in the PBF polymeric chain to ad hoc tune PBF physico-chemical characteristics such as crystallinity and thermal, mechanical, and barrier properties. In addition, being camphoric acid characterized by the presence of a cycloaliphatic moiety, it is supposed to preserve PBF chemical and thermal stability, similarly to what is described in the literature for the *trans*-1,4-cyclohexanedicarboxylic acid-based polymers [37,38,39]. More specifically, the poly(butylene furanoate/camphorate) copolymer family, P(BF_m_BC_n_), where m and n represent the molar percentages of the two co-units, was synthesized by two-step melt polycondensation in the absence of solvents. In order to evaluate the effect of the introduction of a comonomer in the macromolecular chain of PBF, the copolymers were subjected to a complete solid-state characterization after being processed into films by compression molding. Specifically, molecular, thermal, and structural characterizations were carried out. Mechanical and gas barrier properties were also tested, and the results obtained correlated to the copolymer composition.

## 2. Results and Discussion

### 2.1. Synthesis and Molecular Characterization

The copolymers object of the study were synthesized starting from fully bio-based monomers, in detail two different diacid moieties: dimethyl furanoate (DMF), which is the dimethyl ester of 2,5-furandicarboxylic acid (FDCA), and (1R, 3S)-(+)camphoric acid (CA), together with 1,4-butanediol (BD). As one can see in Figure 1a, FDCA contains a five-membered aromatic cycle with two ester groups in the 2,5-position. The COOH-C2-O and O-C5-COOH bond angles are 129.4° [40], and the hybridization of carbons in the ring is sp^2^. Concerning camphoric acid, it contains a five-membered aliphatic cycle with two carboxylic groups in the 1,3-position, two methyl groups on C2, and a further one on C1. Camphoric acid presents a defined configuration at the two stereocenters (1R,3S); the carbons of the ring are hybridized sp^3^, and the COOH-C1-C2 and C2-C3-COOH bond angles are 111.0 and 111.8°, respectively.

As one can see from the 3D representation of the two monomers (Figure 1b), the chemical structure and the molecular geometry are very different, being planar for FDCA while voluminous, bulky, and with a high free volume for CA. In addition, the presence of three methyl groups in CA is expected to limit the crystallization capability of the final polymer [34]. Additionally, the propensity to establish inter-chain interaction is different, being more effective for the planar and aromatic FDCA rings that are able to form not only π-π ring stacking but also hydrogen bonds.

The chemical structure of PBF and P(BF_m_BC_n_) copolymers was confirmed by ^1^H-NMR analysis. For the two copolymers, the chemical composition and the degree of randomness (b) were also evaluated. All the spectra did not reveal any impurities or additional peaks, confirming the expected structures. As an example, in Figure 1, the ^1^H-NMR spectrum of P(BF_70_BC_30_) is shown, together with the assignment of its main peaks. More in detail, together with the signals of chloroform (CHCl_3_) and tetramethylsilane (TMS), used as references, the singlet related to the furan moiety, *a*, can be observed at 7.24 ppm. The signals *b*, *j* (triplets), *c*, and *k* (multiplets), related to the butylene subunit, arise at 4.40, 4.18, 1.91, and 1.80 ppm, respectively. As to the methylene protons of the camphoric ring, the corresponding *e*, *e′*, *f*, and *f′* signals can be observed at 1.51, 2.20, and 2.58 ppm, while the *g* peak coming from the proton in α position to the carboxylic group is located at 2.79 ppm. Finally, the *d*, *h*, and *i* signals related to the protons of the three methyl groups can be observed in the region between 0.75 and 0.85 ppm and around 1.25 ppm, respectively.

The actual molar composition was determined from the normalized area underlying the *j* signal of the butylene protons near the camphoric moiety and the area of the *b* peak of the butylene protons near the furan ring, resulting very close to the feed one.

In order to calculate the degree of randomness b, the region between 1.65 and 2.05 ppm (Figure 1, magnification), where the butylene protons of the *c* and *k* peaks are located, has been used. As one can see, four different signals can be detected: the *c* peak corresponding to Furan–Butylene–Furan (F-B-F) moieties, the *k* peak arising from Camphor-Butylene-Camphor (C-B-C) combinations, and two additional multiple peaks, *c′* and *k′*, located in between, related to the F–B-C segment. The value of b, which is equal to 1 for random copolymers, 2 for alternate copolymers, and 0 < b < 1 for block copolymers, was calculated as follows:b = P_F-C_ + P_C-F_
where P_F-C_ and P_C-F_ are the probability of finding a camphor subunit next to a furan one and the probability of finding a furan moiety next to a camphor one, respectively, and can be expressed, in turn, according to the following equations:$$P_{F\text{-}C}=\frac{I_{F\text{-}B\text{-}C}}{I_{F\text{-}B\text{-}C}+I_{F\text{-}B\text{-}F}};\;P_{C\text{-}F}=\frac{I_{C\text{-}B\text{-}F}}{I_{C\text{-}B\text{-}F}+I_{C\text{-}B\text{-}C}}$$

For both copolymers, *b* values are close to 1 (Table 1), confirming the random distribution of the co-units along the macromolecular chain. The molecular weight was determined by GPC. According to the data reported in Table 1, the synthetic procedure adopted permitted the obtaining of polymers of high and comparable molecular weights and polydispersity indexes (Ð) between 2.2 and 2.6, in line with those obtained by polycondensation. Thus, good control over the synthetic procedure was confirmed. This result is of particular interest, considering that in a previous work from the authors, in which camphoric acid has been copolymerized with *trans*-1,4-cyclohexanedicarboxylic acid, the introduction of only 10 mol% of the camphoric subunit determined a lowering of the copolymer molecular weight with respect to the parent homopolymer [36]. Indeed, the steric hindrance of the methyl group on the carbon in α position to the carboxylic group of the camphoric moiety renders the electrophilic carboxylic carbon less accessible to the nucleophilic attack of the hydroxyl group of the butanediol. For the materials under study, this limitation was overcome, thanks to an optimization of the polymerization process.

### 2.2. Surface, Thermal and Structural Characterization

Static water contact angle (WCA) measurements were performed on compression-moulded films to obtain information about surface wettability. WCA values are listed in Table 1. PBF has a WCA value of 90°. As one can see, the introduction of 10 mol% of camphoric acid did not alter the wettability of the surface, while in the case of a higher amount of co-unit, the hydrophobicity is increased, with WCA reaching a value of 97° in the P(BF_70_BC_30_) copolymer.

Thermogravimetric analysis (TGA) was performed on purified films to determine which temperatures must not be exceeded during processing. T_onset_ and T_max_ data for all the materials under study are listed in Table 2, while the relative thermograms are shown in Figure 2A. All the polymers are characterized by excellent thermal stability, with T_onset_ always above 370 °C. Degradation occurs in two steps in all cases, the greater one at lower temperatures, and the less intense one starting after 400 °C. PBF is the most stable polymer among the family; its T_onset_ and T_max_ values are in line with previous studies [41,42] and decrease by copolymerization by increasing the amount of camphoric moiety. This behavior indicates that the camphoric ring is more prone to thermal cleavage than the furan one, as expected; i.e., aromatic polymers typically have higher thermal stability than aliphatic ones [43]. In addition, it has already been observed in the literature and by the authors too that furan-based polyesters are characterized by particularly high thermal stability, also due to the formation of hydrogen bonding between furan rings [44,45,46,47,48]. T_onset_ values slightly and regularly decrease with the composition. On the contrary, T_max_ decreases by about 15 °C in the P(BF_90_BC_10_) copolymer with respect to PBF then remains almost constant, as confirmed by the thermogram derivatives shown in Figure 2B. As to the residual char, in all cases a complete degradation occurs, with a weight loss of 100% when a temperature of about 600 °C is reached (Figure 2). In conclusion, it can be assessed that copolymerization did not particularly worsen the excellent thermal stability of the PBF, which is one of its strong points.

The purified powders and the corresponding compression-moulded films were subjected to calorimetric measurements by DSC. The scans were performed, in the case of films, after three weeks of storage at room temperature. This time was necessary for films to uniformize their thermal history, since copolymers are characterized by a T_g_ close to room temperature (Table 2). The DSC curves of powders and films are reported in Figure 3, while the calorimetric data are listed in Table 2.

The powders (Figure 3A) show the typical behavior of semicrystalline materials, presenting at lower temperatures the endothermic phenomenon associated with the glass-to-rubber transition and at higher temperatures the melting of the crystalline phase. Regarding the glass transition phenomenon, in the case of powders, it is quite broad, thus preventing a proper T_g_ and Δc_p_ value determination. Concerning the melting process, the introduction of camphoric moieties in the PBF main chain is responsible for a reduction of both T_m_ and ΔH_m_, indicating the formation of a less perfect and less abundant crystalline phase, respectively, as a consequence of the progressive reduction of the crystallization capability of BF segments. Moreover, in the copolymers, some minor endothermic phenomena, located at about 80 °C, can be observed, indicating the presence of an ordered phase with a low degree of perfection. The WAXS analysis, which will be described in the following, will help clarify this point.

Since a semicrystalline material usually shows a different behavior compared to the same material in a complete amorphous state (i.e., crystals act as physical cross-links, causing an increase in T_g_ values), all the polymers were subjected to a fast cooling from the melt. This thermal treatment aims to obtain amorphous samples. The relative DSC curves (II scan) and the thermal characterization data are shown in Figure 3B and Table 2, respectively. A progressive decrease of T_g_ was observed, together with an increase of the amorphous phase fraction, i.e., higher Δc_p_ values. This effect, which has already been observed in the literature for copolymeric systems containing a camphoric subunit [31,36], confirms that copolymerization is responsible for improved flexibility. As to crystallization capability, in all cases, once exceeding T_g_, an exothermic peak followed by an endothermic one at a higher temperature are detected, with comparable underlying areas (ΔH_cc_ = ΔH_m_), demonstrating that macromolecular chains are able to rearrange into an ordered structure upon heating and confirming that the thermal treatment applied was effective to obtain complete amorphous materials. This crystallization/melting phenomenon is more intense for PBF and decreases with increasing the amount of the camphoric moiety.

In order to analyze the crystalline structure deeply, WAXS analysis was performed (Figure 4A) on powders. The peak positions and the profile shape for all three samples indicate the presence of the phase previously reported by J. Zhu et al. for 2,5-PBF [49] as a unique crystalline component; the only difference is the bigger peak-width in the copolymers, which increases with the amount of camphoric co-units. An estimation of the crystal size, through the coherence length perpendicular to the planes *0 1 0*, can be obtained by the Scherrer formula (Table 2) and reveals the worsening of domain perfection as camphoric units are increased. Moreover, as one can see from Figure 4A, the background baseline in the 2-theta interval between 15 and 30° rises as a consequence of more abundant amorphous domains. Finally, the trend of crystallinity values calculated from WAXS scans that are within the interval 28–37% (Table 2) agrees very well with DSC data. As one can notice, the X_c_ value for P(BF_70_BC_30_) is slightly higher than P(BF_90_BC_10_), despite the higher counit content in P(BF_70_BC_30_). The higher X_c_ value for the copolymer containing 30 mol% of counits could be due to the higher amount of mesophase in P(BF_70_BC_30_). Mesophases could reveal broad peaks at WAXS [38], possibly overlapping the crystalline phase peaks. As a consequence, the calculated X_c_ value could also contain the contribution of mesophase signals.

As to the I-scan DSC traces of films (Figure 3C), in all cases the endothermic baseline deviation, due to the glass-to-rubber transition, can be more easily identified. It is followed by an exothermic peak between T_g_ and T_m_ and an endothermic one at higher temperatures. However, a difference between the homopolymer and the copolymers in the crystallization and melting enthalpies can be observed: PBF purified film can be considered slightly semicrystalline (ΔH_cc_ < ΔH_m_), in line with previous studies carried out also by the authors [41,42]. Conversely, the copolymers can be considered amorphous (ΔH_cc_ = ΔH_m_), confirming the hindering of the crystallization capability of PBF chains due to the insertion of camphoric moieties. Interestingly, the copolymeric compression-moulded films, although amorphous and with T_g_ values close to room temperature, can be easily handled and further characterized. The II DSC scans of films are practically the same as the powders after melt quenching.

WAXS scans of compression-moulded films, reported in Figure 4B, show, in all cases, bell-shaped patterns typical of amorphous materials, in agreement with DSC data.

### 2.3. Stress-Strain Measurements

To evaluate the mechanical properties of the polymeric films under investigation, tensile tests were carried out by measuring the stress as a function of the deformation applied. The values of elastic modulus (E), stress at break (σ_B_), and strain at break (ε_B_) are listed in Table 3, while the relative stress-strain curves are reported in Figure 5.

The results obtained can be generally explained on the basis of molecular weight, chain flexibility (i.e., T_g_ value), and crystallinity degree, which are the main parameters affecting mechanical response. According to the data collected in Table 3, PBF turned out to be the most brittle material with the highest value of elastic modulus (E = 1190 MPa) and the lowest elongation at break (ε_B_ = 157%), although not negligible. This behavior can be ascribed to the slight semicrystalline nature of the homopolymer, together with its T_g_ above room temperature, in agreement with previous studies [41,42]. Conversely, the mechanical response of the copolymers is different, with E decreasing by about one third in P(BF_90_BC_10_) and about 30 times in P(BF_70_BC_30_), and ε_B_ significantly improving, reaching about 700% in the copolymer richest in camphoric moieties. This trend cannot be related to the crystallinity degree, which is almost negligible and comparable in all cases, but can be ascribed to the progressive reduction of the T_g_ in the copolymers with respect to the homopolymer, this last indicating higher flexibility of the macromolecular chains. As to the values of σ_B_, a different behaviour of the two copolymers, with respect to PBF, can be observed: in P(BF_70_BC_30_) a decrease of about one order of magnitude was observed, while it increased in P(BF_90_BC_10_) (21 vs. 31 MPa), indicating a material characterized by a strong mechanical resistance. In addition, yielding is present at an elongation at break at about 2% for PBF, becomes less visible in P(BF_90_BC_10_) and completely disappears in P(BF_70_BC_30_).

These results highlight how powerful the introduction of different amounts of camphoric moieties in the PBF macromolecular chain is in tuning the mechanical response of the final materials. Indeed, among the family under investigation, P(BF_90_BC_10_) is the best compromise between high mechanical resistance, considerable ductility, and remarkable flexibility, while P(BF_70_BC_30_) can be considered a thermoplastic elastomer. This is particularly important in the field of food packaging, where different materials are required depending on the kind of packaged food and the storage conditions.

### 2.4. Gas Barrier Properties Measurements

In view of applications in the field of food packaging, permeability measurements are of extreme importance to evaluate how a material can act as a barrier against the external environment, preserving and even elongating the shelf life of foods. To this end, gas barrier properties to O_2_ and CO_2_ were evaluated, and the relative GTR values are shown in Table 3 and Figure 6. In Figure 6, GTR values of PET and PEF are also reported [50,51,52], with the former being the most common plastic used for food and beverage packaging purposes and the latter being the most investigated furan-based material for the same applications.

As to PBF, it is characterized by very good barrier properties, comparable to those of PEF and better than those of PET. As reported in the literature and confirmed by previous studies from the authors, considering that furan moieties are an example of mesogenic groups, the reason for the high performance of furan-based polyester can be explained on the basis of the development of a 1D or 2D-ordered structure, called mesophase, different from the conventional crystalline one. This peculiar order comes from the formation of hydrogen bonds between adjacent chains as well as interplanar π-π stacking of the rings [42,47,48,53,54]. These interactions lead to the formation of a compact and dense net that is particularly efficient in blocking gas passage through the polymeric film, even more so than the crystalline phase.

The copolymers are characterized by truly exceptional gas blocking capability, i.e., gas transmission rate (GTR) values two orders of magnitude lower than PBF to both gases (Figure 6), which regularly decrease as the amount of BC co-units increases. As evidenced by Figure 6, the oxygen barrier capability of the copolymers under study is comparable to that of ethylene vinyl alcohol (EVOH), a polymer largely used in multi-layered, high gas barrier films for food packaging. EVOH presents outstanding O_2_ barrier properties, but very poor mechanical resistance [55]. The exceptional improvement of PBF barrier performance by modification with camphoric acid is exceedingly surprising if we consider that in the most performant material, P(BF_70_BC_30_) copolymer, only 30 mol% of furan moieties have been replaced by camphoric ones. Besides furan content reduction, chemical modification also produces a decrement in T_g_ values. As commonly accepted, lower T_g_ values imply an increase in free volume, which is responsible for an easier passage of gas through the polymeric matrix and thus a worsening of the barrier’s performance. However, in the case of mesogenic materials, such as polyesters based on 2,5-FDCA, the formation of mesophase is particularly favored when T_g_ is close to T_room_ [41,42,56,57,58]. Thus, in the two copolymers under study, this effect could prevail over the lower density of furan moieties per chain length, which are more effective in blocking gas passage than camphoric ones. The establishment of π-π and hydrogen inter-chain interactions could also help explain why the copolymeric films, although amorphous and almost rubbery at room temperature, can be easily handled. On the other hand, another possible reason for the increased gas barrier properties in the copolymers can be found in the chemical structure of camphoric rings, which could further limit gas passage. Indeed, the conformationally rigid five-membered aliphatic ring, the presence of a -CH_3_ group on C1, and the other two substituent methyl groups on C2, could hinder the single bond rotation, enhancing macromolecular rigidity. That could limit gas passage through the polymeric matrix, as the path that gas molecules should follow to cross the polymeric film is particularly tortuous.

## 3. Materials and Methods

### 3.1. Materials

Dimethyl 2,5-furanoate (DMF) was purchased from Sarchem Labs, (1R,3S)-(+)-Camphoric acid (CA) was purchased from J&K Scientific, while 1,4-butanediol (BD), titanium tetrabutoxide (TBT) and titanium isopropoxide (TIP) were purchased from Merck. All reagents were used without any further purification.

### 3.2. Synthesis

Poly(butylene furanoate) (PBF) was synthesized from DMF and BD, while poly(butylene furanoate/camphorate) P(BF_m_BC_n_) random copolymers were prepared using different DMF/CA ratios as diacid moieties and BD as a glycolic monomer (Figure 2). In all cases, a 100% molar excess of glycol was used with respect to diacid/diester content. All the reactions were carried out in bulk, in a 250 mL glass reactor with a thermostated silicon oil bath, using TBT and TIP as catalysts (about 200 ppm of each catalyst). The copolyesters, as well as the PBF homopolymer synthesized as a reference, were obtained according to the usual two-stage polymerization procedure. More in detail, the first stage was carried out under pure nitrogen flow; the temperature was set at 180 °C and kept constant until more than 90% of the theoretical amount of water and methanol was distilled off (about 90 min), as a consequence of esterification and transesterification reactions, respectively. During the second stage, the pressure was progressively lowered to 0.06 mbar while the temperature was increased to 230 °C to favor the removal of the glycolic excess and promote polycondensation. The reactions were stopped once a constant torque value was reached (about 150 additional minutes).

After the synthesis, the so-obtained polymers, which were yellow colored, were purified by dissolution in chloroform containing 5% of 1,1,1,3,3,3-hexafluoro-2-propanol and further precipitation in methanol in order to remove unreacted monomers and catalysts. After purification, the samples appeared like small flakes, lighter in color.

### 3.3. Film Preparation

Films of about 10 cm in diameter and a thickness of 150 μm were obtained by compression molding (Carver C12, laboratory press) the purified polymers between two teflon squared shits at a temperature of 30 °C higher than their melting point. After complete melting, a pressure of about 4 ton/m^2^ was applied for 2 min. Then, the films were ballistically cooled to room temperature in the press. Prior to characterization, all the films were stored under vacuum at room temperature for three weeks in order to reach thermal equilibrium.

### 3.4. Molecular Characterization

Both the homopolymer and the copolymers were characterized from the molecular point of view by means of proton nuclear magnetic resonance spectroscopy (^1^H-NMR) in order to confirm their chemical structure and, in the case of the copolymers, their chemical composition and architecture. The samples were first dissolved in a mixture (80:20 *v*:*v*) of chloroform-d containing 0.03 v% tetramethylsilane (TMS), as an internal standard, and trifluoroacetic acid. The measurements were carried out at room temperature, employing a Varian Inova 400-MHz instrument (Agilent Technologies, Santa Clara, CA, USA). A relaxation delay of 0 s, an acquisition time of 1 s, and up to 100 repetitions were employed.

Molecular weight (M_n_) and polydispersity index (Ð) were measured by gel-permeation chromatography (GPC) at 30 °C using a 1100 HPLC system (Agilent Technologies) equipped with a PLgel 5-μm MiniMIX-C column (Agilent Technologies) and a refractive index detector. A mixture of chloroform containing 5% of 1,1,1,3,3,3-hexafluoro-2-propanol was used as an eluent with a 0.3 mL/min flow. The samples were first solved at a concentration of about 2 mg/mL in the same solvent used as an eluent and then injected. To obtain the molecular weights starting from the elution times, a calibration curve obtained with polystyrene standards in the range of 2000–100,000 g/mol was used.

### 3.5. Surface, Thermal and Structural Characterization

The surface wettability of each compression moulded film was evaluated by static water contact angle (WCA) measurements by means of a KSV CAM101 instrument and the software Drop Shape Analysis. The WCA value, reported as the mean value ± standard deviation of at least 10 measurements, was obtained by the side profile of each deionized water drop immediately after deposition on the polymeric surface.

In order to determine the temperature of initial degradation (T_id_) of the samples and the one at which degradation occurs at maximum rate (T_max_), thermogravimetric analysis (TGA) was carried out under nitrogen atmosphere using a Perkin Elmer TGA7 apparatus. The measurements were performed under a nitrogen flow of 40 mL/min, by heating weighed samples of about 10 mg at a rate of 10 °C/min in the temperature range of 40–800 °C.

Calorimetric measurements were performed by means of a Perkin Elmer DSC7 instrument under pure nitrogen flux. By using this technique, it was possible to calculate the glass-transition temperature (T_g_), taken as the midpoint of the heat capacity increment Δc_p_ associated with the glass-to-rubber transition of the amorphous phase, which was, in turn, calculated from the distance between the two extrapolated baselines at T_g_. The cold crystallization (T_cc_) and the melting temperatures (T_m_) were instead determined as the peak values of the exotherms and endotherms in the DSC curve, respectively. The heat of cold crystallization (ΔH_cc_) and the heat of fusion (ΔH_m_) of the crystalline phase were calculated from the total areas of the relative exothermal and endothermal phenomena, respectively.

Wide-angle X-ray scattering (WAXS) was performed by means of an X’Pert PANalytical diffractometer equipped with a fast X’Celerator detector, operating in reflection mode and using the wavelength of the K_α_ radiation of copper (λ = 15418 Å). The 2θ interval from 5° to 60° was explored (step of 0.1° and counting time of 100 s/point). The crystallinity degree (X_c_) was calculated as the ratio between the crystalline diffraction area (A_c_) and the whole area of the diffraction profile (A_t_). The crystalline diffraction area was obtained, in turn, from the total area of the diffraction profile by subtracting the amorphous halo, which was modeled as a bell-shaped peak. The incoherent scattering was also taken into account. The length of coherent domains perpendicular to a specific plane direction was calculated by the Scherrer equation [59].

### 3.6. Stress-Strain Measurements

The tensile testing of PBF and its random copolymers was performed using an Instron 5966 dynamometer, equipped with a rubber grip and a transducer-coupled 10 kN loading cell controlled by software. Each rectangular film (5 × 50 mm) was fixed to the instrument with an initial gauge length of 20 mm and then subjected to elongation at a constant speed of 10 mm/min until break. In real time, a stress-strain curve was obtained from the software. By these measurements, it was possible to calculate the elastic modulus (E), as the slope of the initial linear part of the obtained curve, the elongation (ε_B_) and stress at break (σ_B_), which were considered the maximum elongation and stress values reached by the sample before breaking. At least six different samples were tested for each polymer, and the results were provided as the average value ± standard deviation.

### 3.7. Gas Barrier Properties Measurements

The evaluation of the gas barrier performance was carried out by a manometric method using a Permeance Testing Device, type GDP-C (Brugger Feinmechanik, München, Germany), according to ASTM 1434-82 (Standard Test Method for Determining Gas Permeability Characteristics of Plastic Film and Sheeting), DIN 53 536 in compliance with ISO/DIS 15 105-1, and according to the Gas Permeability Testing Manual (Registergericht München HRB 77020, Brugger Feinmechanik GmbH, Munich, Germany).

All the measurements were performed at a temperature of 23 °C with a gas stream of 100 cm^3^·min^−1^. The gases tested, oxygen and carbon dioxide, were food-grade at 0% RH. Samples with an area of 78.5 cm^2^ were first placed in the instrument, between the upper and lower analysis chambers, and a preliminary high vacuum desorption of both chambers was applied. Then, the upper chamber was filled with the tested gas at ambient pressure, and a pressure transducer set in the lower chamber continuously recorded the increasing gas pressure with time. The gas transmission rate (GTR) measurements were performed at least in triplicate by determining the increase in pressure in relation to time, the volume of the device, and the thickness of the film. This last value was obtained by a digital micrometer as the mean value of three experimental tests performed at ten different points on the film surface at room temperature. GTR values were provided as the average value ± standard deviation.

## 4. Conclusions

Poly(butylene 2,5-furandicarboxylate) was successfully copolymerized with camphor, a highly available, cheap, and renewable building block, to obtain two random, fully bio-based copolymers with 10 and 30 mol% of co-unit, respectively. Thanks to the chemical modification proposed, the rigidity of PBF was successfully decreased without compromising its high thermal stability and further improving its already good functional properties (mechanical as well as gas barrier properties), such as those requested in the field of flexible food packaging. Thus, the new family of copolymers revealed themselves to be particularly promising for the realization of high-performance flexible and mono-layered food packaging films.

More in detail, as evidenced by calorimetric measurements, copolymerization allowed to increase the chain flexibility of PBF, making possible the realization of flexible amorphous thin films with a T_g_ around room temperature, whose mechanical characteristics can be tailored depending on the amount of camphoric moieties present in the final material. For all the copolymers investigated, stress-strain measurements indicated the introduction of BC moieties is responsible for a progressive decrease of Young’s modulus, together with a parallel improvement of the elongation at break. As to the mechanical resistance, it changed according to the content of camphoric acid. Indeed, for the smallest amount of BC co-units, stress at break increased with respect to PBF, while in the other case, σ_B_ resulted in a lower value than the value measured for the PBF parent homopolymer. The film of the copolymer containing the highest amount of camphoric acid exhibited the typical behaviour of thermoplastic elastomers. Thus, interestingly, just playing on copolymer composition, we could obtain two materials both characterized by high ductility, but whose resistance and toughness are high or low according to a lower or higher BC co-unit content, respectively.

Most importantly, copolymerization deeply and positively affected gas barrier properties, the extent of improvement being strictly proportional to the amount of camphoric acid introduced along the PBF macromolecular chain, reaching the performance of EVOH, a polymer largely used in food packaging for its outstanding gas barrier capability but combined with other polymers, usually polyolefins, in multi-layered films to overcome its extremely poor mechanical resistance.

Additionally, for these new materials, the outstanding mechanical and gas barrier properties can be correlated to the presence of a phase arising from π-π hydrogen inter-chain interactions and similar to the mesophases present in polymeric liquid crystals, whose formation is maximized when the polymer’s T_g_ ≈ T_room_.

Considering the high barrier performance as well as the mechanical resistance, it can be assessed that this family of bio-based polyesters proved to be very promising and suitable for the realization of a high barrier monolayer packaging with tunable mechanical behavior and reduced costs of production.

## Data Availability

Data available on request due to privacy.

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
