# Peer review of "New Random Aromatic/Aliphatic Copolymers of 2,5-Furandicarboxylic and Camphoric Acids with Tunable Mechanical Properties and Exceptional Gas Barrier Capability for Sustainable Mono-Layered Food Packaging"

_molecules, 2023, doi:10.3390/molecules28104056_

Round 1

Reviewer 1 Report

The article “New random aromatic/aliphatic copolymers of 2,5-furandicar- boxylic and camphoric acids with tunable mechanical proper- ties and exceptional gas barrier capability for sustainable mono-layered food packaging”, submitted by G. Guidotti, M. Soccio, M. Gazzano, V. Siracusa and N. Lotti, describes the synthesis and the structural characterization of biobased polyesters for food packaging application. The applicability was further confirmed by evaluating the mechanical and gas barrier properties of the films produced with those polymers. 

In my opinion, the document is quite comprehensive and well-written. Perhaps, considering the intended application for the materials, to conclude the typical film characterization, contact angle measurements should be added, if the technique is available. 

I suggest publishing the article after some minor revisions, such as the following:

In line 16 the camphoric acid is described as “(1R, 3S)-(+)-Camphoric Acid (CPA)”, in line 77, and subsequently throughout the text, as “(1R, 3S)-(+)-Camphoric Acid (CA)”; please maintain the correct one.

Line 59: “monomer that caught” could be better.

Line 85 and others' “interest in” could be better.

Lines 261 – 262 comments should be removed.

Line 59: “monomer that caught” could be better.

Line 85 and others' “interest in” could be better.

Author Response

The article “New random aromatic/aliphatic copolymers of 2,5-furandicar- boxylic and camphoric acids with tunable mechanical proper- ties and exceptional gas barrier capability for sustainable mono-layered food packaging”, submitted by G. Guidotti, M. Soccio, M. Gazzano, V. Siracusa and N. Lotti, describes the synthesis and the structural characterization of biobased polyesters for food packaging application. The applicability was further confirmed by evaluating the mechanical and gas barrier properties of the films produced with those polymers. 

In my opinion, the document is quite comprehensive and well-written. Perhaps, considering the intended application for the materials, to conclude the typical film characterization, contact angle measurements should be added, if the technique is available. 

We thank the reviewer for his/her kind suggestion. We added the water contact angle data and discussed them in the text.

I suggest publishing the article after some minor revisions, such as the following:

In line 16 the camphoric acid is described as “(1R, 3S)-(+)-Camphoric Acid (CPA)”, in line 77, and subsequently throughout the text, as “(1R, 3S)-(+)-Camphoric Acid (CA)”; please maintain the correct one.

We thank the reviewer for his/her correct observation. We changed the abbreviation in line 16 accordingly.

Line 59: “monomer that caught” could be better.

We thank the reviewer for his/her correct observation. We changed the expression in line 59 accordingly.

Line 85 and others' “interest in” could be better.

We thank the reviewer for his/her correct observation. We changed the expression in lines 65 and 85 accordingly.

Lines 261 – 262 comments should be removed.

We thank the reviewer for his/her correct observation. We removed the comment in the text.

Reviewer 2 Report

Comments to the Author

This manuscript synthesized the aromatic/aliphatic copolymers containing 2,5-furandicar-2-boxylic and camphoric acids units and analyzed their thermal, mechanical, rheological, and gas barrier properties. The analysis of new aromatic/aliphatic copolymers is sufficient, and the English writing is easy to read. It has various flaws which need to be taken care of before publication.

1.          On page 11, line 336, “As commonly accepted, lower Tg values imply an increase in free volume, which is responsible of an easier passage of gas through the polymeric matrix, and thus a worsening of the barrier performance. However, in case of mesogenic materials, such as polyesters based on 2,5-FDCA, the formation of mesophase is particularly favored when Tg is close to T room [41,42,56-58].” The reviewer can accept this interpretation partially. The authors can try to explain from the point of view of the stereo hindrance and twisted structure of CA to enhance the gas barrier. What do the authors think about these points? Hopefully, this view can help the reader understand the role of CA within the copolyester.

2.          The reviewer is curious, what the characteristic of PBC (100% CA) is? From the DSC result, the Tm may locate at ~100 or become fully amorphous materials, but the mechanical will weaken. Furthermore, the reviewer thinks the PBC has potential use in the gas barrier packaging materials at a lower temperature for frozen food how the author thinks about this point. Does the PBC hard to synthesize?

3.          In Table 2, why does the Xc increase when the CA increases to 30 mole%(Xc=32%) from powder WAXS results?

4.          In Figure 2, the TGA curve shows two stages of degradation, and the reviewer suggests the DTG can be provided the discussed. Furthermore, the curve of P(BF70BC30) is slightly above 100% at a temperature below 200 ; the author can check this point.

5.          The reviewer is interested in why the authors use a mixture solvent of chloroform containing 5% of 1,1,1,3,3,3-hexafluoro-2-propanol for GPC measurement. Why not adopt 100% HFIP?

6.          What does the role of 3 CH3 groups of the camphoric acid act in the gas barrier?

English writing is easy to read.

Author Response

This manuscript synthesized the aromatic/aliphatic copolymers containing 2,5-furandicar-2-boxylic and camphoric acids units and analyzed their thermal, mechanical, rheological, and gas barrier properties. The analysis of new aromatic/aliphatic copolymers is sufficient, and the English writing is easy to read. It has various flaws which need to be taken care of before publication.

  1. On page 11, line 336, “As commonly accepted, lower Tg values imply an increase in free volume, which is responsible of an easier passage of gas through the polymeric matrix, and thus a worsening of the barrier performance. However, in case of mesogenic materials, such as polyesters based on 2,5-FDCA, the formation of mesophase is particularly favored when Tg is close to T room [41,42,56-58].” The reviewer can accept this interpretation partially. The authors can try to explain from the point of view of the stereo hindrance and twisted structure of CA to enhance the gas barrier. What do the authors think about these points? Hopefully, this view can help the reader understand the role of CA within the copolyester.

We thank the reviewer for the opportunity to clarify this point. In the paper we assessed that the main reason of the better performance of the copolymers with respect to PBF, directly proportional to the amount of co-unit, was the decrease of Tg towards values near room temperature. Indeed, according to the literature, mesophase amount is maximized under this condition. However, the rigidity of CA cannot be excluded when trying to explain the barrier behavior of the copolymer, as suggested by the referee. Indeed, CA consists of a conformationally rigid five-membered aliphatic ring, with a -CH3 group on C1, and other two substituent methyl groups on C2. This peculiar structure hinders the single bonds rotation enhancing the macromolecular rigidity.  That could limit gas passage through the polymeric matrix, making the path that gas molecules should follow to cross the polymeric film very tortuous. We have modified the text accordingly.

  1. The reviewer is curious, what the characteristic of PBC (100% CA) is? From the DSC result, the Tm may locate at ~100 ℃or become fully amorphous materials, but the mechanical will weaken. Furthermore, the reviewer thinks the PBC has potential use in the gas barrier packaging materials at a lower temperature for frozen food how the author thinks about this point. Does the PBC hard to synthesize?

We thank the reviewer for the opportunity to clarify this point. We managed to synthesize the homopolymer PBC by the same synthetic route used for PBF and the copolymers. Unluckily, the material turned out to be completely amorphous with a rubbery amorphous phase (Tg = 12 °C). Thus, it was impossible to obtain a free-standing film by compression molding and to perform further functional characterization, apart from thermal one. As suggested by the referee, the possible use of PBC homopolymer for the packaging of frozen food could be a possibility, but we must take into account that after molding, the storage temperature must never exceed its Tg, or it will become not manageable. Thus, this is a not-negligible limit in the use of the material.

  1. In Table 2, why does the Xc increase when the CA increases to 30 mole%(Xc=32%) from powder WAXS results?

We thank the reviewer for his/her question. The slightly higher Xc value for the copolymer containing 30 mol% of counits with respect to the other copolymer containing 10 mol% of counits, could be due to higher amount of mesophase in P(BF70BC30). Mesophases could reveal as broad peaks at WAXS (Guidotti et al. Polymers 2018, 10, 866; doi:10.3390/polym10080866) possibly overlapping the crystalline phase peaks. As a consequence, the calculated Xc value could also contain the contribution of mesophase signals.

We have modified the text to clarify the point.

  1. In Figure 2, the TGA curve shows two stages of degradation, and the reviewer suggests the DTG can be provided the discussed. Furthermore, the curve of P(BF70BC30) is slightly above 100% at a temperature below 200 ℃; the author can check this point.

We thank the reviewer for his/her suggestion. We added in Figure 2 the thermogram derivatives for each polymer and we discussed them in the text. We also checked again the TGA curve of P(BF70BC30).

  1. The reviewer is interested in why the authors use a mixture solvent of chloroform containing 5% of 1,1,1,3,3,3-hexafluoro-2-propanol for GPC measurement. Why not adopt 100% HFIP?

We thank the reviewer for the opportunity to clarify this point. We used the mixture of chloroform containing 5% of 1,1,1,3,3,3-hexafluoro-2-propanol instead of pure HFIP due to the fact that this last is not the ideal solvent to elute in the GPC column. Indeed, the column could be damaged by a too aggressive solvent such as HFIP. Moreover, the chloroform used was HPLC grade, specific for this kind of measurement.

  1. What does the role of 3 CH3groups of the camphoric acid act in the gas barrier?

We thank the reviewer for the opportunity to clarify again this point. As already explained in point 1, the presence of a -CH3 group on C1 and other two -CH3 groups on C2, together with the presence of a rigid 5-membered ring, hinders the single bonds rotation enhancing the macromolecular rigidity. That could help explaining the better barrier performances of the copolymers with respect to PBF.

Round 2

Reviewer 2 Report

The authors have modified the suggestions carefully and can be accepted in current form.